# Short-Wave Infrared Hyperspectral Image-Based Quality Grading of Dried Laver (*Pyropia* spp.)

**DOI:** 10.3390/foods14030497

**Published:** 2025-02-04

**Authors:** Jong Bong Lee, Yeon Joo Bae, Ga Yeon Kwon, Suk Kyung Sohn, Hyo Rim Lee, Hyeong Jun Kim, Min Jae Kim, Ha Eun Park, Kil Bo Shim

**Affiliations:** Department of Food Science and Technology, Pukyong National University, Busan 48513, Republic of Korea; whdqhd11@pukyong.ac.kr (J.B.L.); yeonjoo4459@gmail.com (Y.J.B.); gayeon7401@gmail.com (G.Y.K.); jhshon2@naver.com (S.K.S.); lhr2008@pukyong.ac.kr (H.R.L.); hyeongjun5767@gmail.com (H.J.K.); rlaalswo4992@naver.com (M.J.K.); phe5765@naver.com (H.E.P.)

**Keywords:** hyperspectral imaging, quality grading, dried laver, partial least-squares discriminant analysis

## Abstract

Laver (*Pyropia* spp.) is a major seaweed that is cultivated and consumed globally. Although quality standards for laver products have been established, traditional physicochemical analyses and sensory evaluations have notable drawbacks regarding rapid-quality inspection. Not all relevant physicochemical quality indices, such as texture, are typically evaluated. Therefore, in this study, we investigated the use of hyperspectral imaging to rapidly, accurately, and objectively determine the quality of dried laver. Hyperspectral images of 25 dried laver samples were captured in the short-wave infrared range from 980 to 2576 nm to assess their moisture, protein content, cutting stress, and other key quality indicators. Spectral signatures were analyzed using partial least-squares discriminant analysis (PLS-DA) to correlate the spectral data with three primary quality index values. The performance of PLS-DA was compared with that of the variable importance in projection score and nonlinear regression analysis methods. The comprehensive quality grading model demonstrated accuracies ranging from 96 to 100%, R^2^ values from 75 to 92%, and root-mean-square errors from 0.14 to 0.25. These results suggest that the PLS-DA regression model shows great potential for the multivariate analysis of hyperspectral images, serving as an effective quality grading system for dried laver.

## 1. Introduction

Laver (*Pyropia* spp.; phylum: Rhodophyta; class: Bangiophyceae; order: Bangiales; family: Bangiaceae) is a major seaweed cultivated and consumed worldwide, particularly in South Korea, Japan, and China [1]. Known for its high protein and low lipid content [2], laver is rich in essential amino acids, including isoleucine, leucine, valine, and threonine, as well as minerals, such as potassium, phosphorus, and magnesium [3]. Porphyran and phycobiliproteins, the characteristic carbohydrates and proteins in laver, respectively, exhibit antioxidant properties [4,5]. These health benefits have led to a global increase in laver consumption [6]. As the consumption of laver continues to rise because of its health benefits, ensuring its consistent quality has become increasingly important. This need for quality grading is further complicated by the variety of factors that influence its quality, including nutrient composition, texture, and flavor.

After harvesting, raw laver is quickly processed into dried laver through a drying process to prevent quality degradation caused by the pigment (chlorophyll and phycoerythrin) destruction and spoilage [7]. Dried laver is roasted and eaten as is or used as an ingredient in Korean-style rice rolls. It is also used in various products, such as seasonings and snacks. The quality and taste of dried laver, which influence its price, depend on numerous factors, such as nutrient composition, color, flavor, and thickness [8]. Standards for laver products, including the Codex regional standard (CXS 323R-2017 [9]) and national standards (Korea Industrial Standard [KS H 6025] [10] and China National Standard [GB/T 23597-2022] [11]), have been established to ensure their quality. However, other physicochemical quality indices, such as texture, are either not included in these standards or lack objectivity.

The nutritional content and processing characteristics of raw laver can change depending on the time of year it is harvested. Freshly collected laver each year has a high protein content and is highly nutritious; however, this quality tends to decline as production continues. Therefore, it is crucial to establish an objective quality grading system for dried laver to maintain its nutritional value. This system will also help prevent deterioration after processing, such as microbial contamination caused by high residual moisture or damage to the pigments in dried laver [12,13].

The correlation between physicochemical properties and the quality of dried laver has been extensively studied. Superior lavers often have a stronger texture and umami flavor [14]. The umami flavor of laver is primarily attributable to the presence of inosine-5′-monophosphate, alanine, glutamine, and aspartic acid [15]. Additionally, total nitrogen, free amino acids, and phycobiliproteins in lavers are positively correlated with flavor and texture; notably, phycobiliproteins and amino acid-based compounds correlate with protein content [16,17,18]. Therefore, the quality of dried laver is likely related to its protein content and texture. Implementing quality standards for dried laver can mitigate international trade issues and boost the laver industry [19]. Objective and complementary quality standards that include physicochemical and sensory indices are needed to advance the laver industry. Traditional methods for physicochemical analysis are complex and expensive, whereas sensory evaluations are subjective and time-consuming, making them unsuitable for rapid-quality inspection [20].

Recent research has focused on developing nondestructive technologies for rapid-quality evaluation [21]. Previous studies have applied nondestructive testing methods to dried laver to measure chemical components, such as proteins, using techniques that include near-infrared (NIR) and Fourier transform infrared spectroscopy [22,23]. However, these spectroscopic methods typically analyze only a limited area where light is directed, restricting the overall analysis range. Additionally, although some studies have utilized hyperspectral imaging (HSI) systems to predict the protein content of dried laver, they have not been performed to comprehensively examine various quality factors of the product [24].

HSI is an emerging technology that is considered one of the most promising techniques for quality evaluation. HSI combines spectroscopic and imaging methodologies into a single system, simultaneously providing spectral and spatial information. This integrated approach allows HSI to identify multiple components within a product and quantify their spatial distribution, enabling the calculation of the compositional gradient within the product.

HSI systems based on chemometrics are gaining traction for such purposes. HSI provides spatial information by simultaneously analyzing various spectral regions [25]. Regions of interest (ROIs) can be selected within the hyperspectral image, allowing for noise reduction and reliable wavelength extraction [26]. HSI has been used to predict chemical compositions and detect microscopic damage, such as bruises or chemical contamination, in plants, including vegetables and fruits, across ultraviolet/visible (VIS) (400–1000 nm), NIR (1000–1700 nm), and short-wave infrared (SWIR; 1000–2500 nm) wavelengths [27,28,29,30]. The SWIR region, in particular, has been used to effectively assess nonvisible indictors of food quality and holds the potential for feature classification [31,32].

SWIR wavelengths offer superior transmission characteristics compared to VIS and NIR wavelengths [33]. As a result, SWIR is suitable for addressing the transmission limitations experienced with VIS and NIR in current food quality assessments. Additionally, SWIR facilitates the analysis of wavelength characteristics based on various physicochemical compositions. Recently, HSI has been used to predict and classify the moisture content of salmon and the freshness of pork using multivariate regression algorithms [34,35].

This study aimed to develop a grading system that evaluates various quality indicators of dried laver. Furthermore, research has shown that HSI technology, utilizing SWIR, is effective for the simultaneous assessment of multiple components in seafood, including dried laver.

In addition, this study explored the potential of the SWIR HSI system for selecting appropriate indicators for quality evaluation and developing a rapid and nondestructive quality grading system for dried laver. Characteristic spectra of dried laver were extracted using isodata clustering methods and classified using a linear regression model (partial least-squares discriminant analysis [PLS-DA]) and a machine learning method (artificial neural network discriminant analysis [ANN-DA]) based on quality index assessments.

## 2. Materials and Methods

### 2.1. Materials

Dried laver samples (*Pyropia* spp.) were purchased from local markets primarily located in the main producing areas along the western coastal regions of Korea (*n* = 25, Appendix A). The collected samples were used as is for HSI measurements and cutting stress and then ground using a mixer for chemical composition analysis. Boric acid (≥99.5%) and 0.1 N hydrochloric acid were obtained from Deajung Chemicals & Metals Co., Ltd. (Siheung, Republic of Korea). Sodium hydroxide (≥97.0%) and sulfuric acid (≥95.0%) were obtained from Duksan Pure Chemicals Co., Ltd. (Ansan, Republic of Korea). The laver powder was stored at 4 °C for further experiments. All chemicals and solvents used in this study were of analytical or high-performance liquid chromatography grade.

### 2.2. Proximate Composition

The proximate composition, including the moisture and crude protein content of the laver, was measured using Association of Official Agricultural Chemists methods [36]. Moisture content was determined by drying samples in Petri dishes in a digital hot air oven (TMF-2000; Tokyo Rikakikai Co., Tokyo, Japan) at 105 °C for approximately 24 h or until a constant weight was achieved. The protein content was determined using the micro-Kjeldahl distillation method (KjelFlex K-360; BÜCHI Labortechnik GmbH, Essen, Germany), following standard procedures, and converted to total protein (N% × 6.25).

### 2.3. Cutting Stress

Cutting stress was measured using the method described by Lu et al. [37], with slight modifications. The cutting stress of dried laver was measured using a texture analyzer (CR-100; Sun Scientific Co., Ltd., Tokyo, Japan) equipped with a 10 kg load cell and a shear blade (height: 9 mm; width: 45 mm; thickness: 0.35 mm). The dried laver was cut to dimensions of 70 × 15 × 1 mm^3^, and cutting stress was measured by inserting the cutting probe at a 60 mm/min rate.

### 2.4. HSI System and Image Acquisition

Hyperspectral images of each sample were collected using an SWIR hyperspectral camera (SWIR, Specim, Spectral Imaging, Ltd., Oulu, Finland) equipped with an OLES lens (OLES 22,5, Xenics Infrared Solutions, Leuven, Belgium) with a focal length of 22.5 units, a field of view (FOV) of 23°, and an aperture F-value of 2.0. The exposure time for each sample was 10 ms, and the distance from the camera was 590 mm. The camera moved at 26.5 mm/s to collect 288 spectral bands (wavelength range: 980–2560 nm; spectral resolution: 5.5 nm). Illumination was provided by halogen lamps (Decostar 51, Osram GmbH, Munich, Germany) with a 20 W power and a 205 lm intensity arranged on either side of the sample at a distance of 500 mm. Data cubes (*n* = 125) were collected using Lumo Scanner software (v2019.535; Specim, Spectral Imaging, Ltd.). A total of 125 data cubes were acquired, with 5 data cubes collected for each sample, from which hyperspectral images were captured from 5 individual layers.

Samples were placed on a whiteboard during scanning for background calibration. The spectrum of dried laver was calibrated using the reflectance of a white standard plate taken simultaneously with the sample to minimize interference factors.

### 2.5. ROI Selection and Processing

During dried laver production, pores can form because of the characteristics of raw laver and the processing facilities. Excluding these pore areas is necessary to calibrate spectra and reduce data noise. In this study, a square ROI with dimensions of 50 × 300 pixel^2^ around the center was selected, and pore areas were classified using isodata classification [38].

Among the common thresholding algorithms, the isodata clustering method stands out for its ability to compute optimal thresholds. This is achieved by calculating the minimum Euclidean distance between random clusters of thresholds at each step. Notably, the isodata classification method is particularly effective for classifying images with porosity [39]. Figure 1 shows the process of excluding pore areas using isodata classification. The maximum number of classes for classifying the pores of dried laver was set at 100. This decision was based on previous experiments, which showed that the variation in pixel values corresponding to the pores remained within 5% after classification. The threshold for this classification was established at 5%, and the process was repeated 10 times to ensure accuracy. Points 1–3 demonstrated changes in the spectrum corresponding to different porosity levels. Image processing and ROI selection were conducted using ENVI v5.7 (NV5 Geospatial Solutions, Inc., Broomfield, CO, USA).

### 2.6. Data Analysis

#### 2.6.1. PLS-DA

PLS-DA maximizes the covariance, Cov (t, Y), between the output variable (Y) and the input variables (X) through a combination of linear separators [40]. Because of these features, PLS-DA has shown successful results in spectral data-based classification [41]. In the present study, each sample spectrum was graded based on the quality index using Tukey’s analysis of variance (ANOVA) to confirm the accuracy and performance of the PLS-DA model. Predictions for the dried laver were carried out based on the trained model.

#### 2.6.2. Wavelength Selection

This study identified important wavelengths using a variable importance in projection (VIP) score based on PLS-DA. The VIP score was defined by the proportion of variance in Y and the PLS weights w (1, 2, 3). The internal weight for each X variable in Y [42] was calculated using the following formula:(1)$${VIP}_{f}^{2}=\sum_{f=1}^{n}\frac{w_{if}^{2}(b_{ff}^{2}t_{f}T_{t_{f}})}{{(b}^{2}T^{T}T)\times F}$$

If the VIP^2^ score was ≥1, the corresponding X variable was classified as a critical wavelength for building the classification model. These critical wavelengths were collected to reconstruct the wavelength for VIP-PLS-DA analysis.

#### 2.6.3. ANN Discriminant Analysis (ANN-DA)

ANN partition data are in a probabilistic form, unlike the PLS model data, which have a mathematical basis. ANN simulates neuron and brain behavior, processing input data through the node of each layer, similar to that in neurons, and outputs probabilistic results to the output layer [43]. In building a nonlinear regression model, such as an ANN, supervision normalizes the results through weight correction for classification according to the resulting supervision information [44]. These features can correct errors that occur in classification models based on mathematical linear regression for data prediction [45]. These trained models were developed through supervised learning, and their performance was examined using a predicted model. To compare the classification performance of ANN-DA, the trained model was supervised using Tukey’s ANOVA, multiple comparison tests, and PLS.

#### 2.6.4. Performance Analysis of the Regression Model

The accuracies of the PLS-DA and ANN-DA classification models were evaluated based on the results of the training and test sets [46]. Evaluation parameters included sensitivity, specificity, accuracy, and error rate and were calculated using the following equations:(2)$$Sensitivity=\frac{TP}{TP+FN}$$(3)$$Specificity=\frac{TN}{TN+FP}$$(4)$$Accuracy=\frac{TP+TN}{TP+TN+FP+FN}$$(5)$$Error\;rate=\frac{FP+FN}{TP+TN+FP+FN}$$
where TP (true positive) represents the number of samples correctly classified as the target class; TN (true negative) represents the number of samples incorrectly classified as the target class; FP (false positive) represents the number of samples correctly classified as a different class; and FN (false negative) represents the number of samples incorrectly classified as a different class. Sensitivity and specificity represent the classification performance, whereas accuracy and error rate indicate the composite performance. Sensitivity is defined as the ability of the model to correctly recognize samples belonging to the target class. The specificity is defined as the ability of the model to correctly recognize samples belonging to a different class [47]. Accuracy is calculated as the arithmetic means of all classes, and the error rate is defined as 1—accuracy. All four regression models, (1) PLS-DA, (2) VIP-PLS-DA, (3) ANN-DA, and (4) PLS-ANN-DA, were run using the PLS toolbox (v9.3; EigenVector, Manson, WA, USA) in MATLAB (R2024a; MathWorks, Natick, MA, USA).

To construct the four quality classification models, 80% of the total of the samples was used to develop a learning model, and 20% was set aside for quality prediction. Cross-validation of the model was performed using the Venetian blind method, with the number of data split equal to 10 [48].

### 2.7. Statistical Analysis and Sample Classification

Statistical analysis of the quality index of dried laver was conducted using SPSS (v27; SPSS Inc., Chicago, IL, USA). ANOVA was used to conduct Tukey’s multiple comparison tests, with statistical significance set at *p* < 0.05. Data were reported as mean ± standard deviation. Samples were divided into three grades based on Tukey’s ANOVA. After confirming classification feasibility, the criteria for crude protein and moisture were applied according to the KS standard (Korea’s industrial standard for dried laver), and cutting stress was applied based on Tukey’s ANOVA to develop a composite quality grading system.

## 3. Results

### 3.1. Quality Index Analysis and Sample Classification

The physicochemical components (moisture, protein, ash, total amino acids, organic acids, and ATP-related compounds) of 25 types of dried laver were analyzed (Appendix A). To evaluate the predictability of the physical and chemical composition of dried laver, PLS-DA was performed using Tukey’s ANOVA on 34 components. The accuracy and R^2^ values ranged from 82 to 90%. This study identified moisture, protein content, and cutting stress as significant quality factors for developing a grading system for dried laver (Appendix A).

The moisture content ranged from 6.07 ± 0.04 to 12.04 ± 0.03 g/100 g, the crude protein content ranged from 32.27 ± 0.28 to 44.99 ± 0.06 g/100 g, and the cutting stress ranged from 136.63 ± 91.67 to 362.10 ± 164.09 kg/cm^2^ (Appendix A). The results of the quality index analysis and sample classification based on Tukey’s ANOVA are shown in Figure 2. Moisture and crude protein were separated into three groups, whereas cutting stress showed significant deviations across samples. This variability was attributable to the features of raw laver and porosity resulting from processing [49,50]. As a result, cutting stress was classified according to PLS-DA based on Tukey’s ANOVA.

### 3.2. Spectral Characteristics

Figure 3 presents the sample spectra extracted from the hyperspectral cubes of dried laver. The first separation occurred between wavelengths of 1400 and 1700 nm, and the second occurred at wavelengths over 1850 nm. These spectral trends depend on the composition of functional groups and shape of the sample [51]. Wavelengths between 1600 and 1800 nm were related to amide I, and those at approximately 1470–1570 nm were related to amide II and the first overtone of O-H in the water band. Amide I bands generally occur after the bonding of the N-H group of a protein to the carboxyl group of an organic substance [52]. Bands at approximately 1950 nm are associated with the O-H in water and the C=O covalent bond of ketonic carbonyl COOH in polysaccharides. The band observed at 2100–2200 nm is related to the aromatic C=C functional group [53]. Finally, the band at 2200–2500 nm has been reported to correlate with the stretching vibration and second overtone of CH [54].

### 3.3. Wavelength Selection for Classifying Quality Indices

Figure 4 shows the critical wavelengths for the classification of each quality index as determined by the VIP score based on PLS-DA classification. Trends of the critical wavelengths are consistent with the functional groups observed in the spectra of the sample. The moisture classification is mainly related to the ranges of 1396–1468 nm, 1885–1984 nm, and 2543–2576 nm. These ranges match those of O-H bands. For crude protein, the critical ranges observed are 1691–1746 nm, 1896–2032 nm, and 2455–2576 nm. These ranges are related to the amide I, carboxyl, and functional groups of amino acids, respectively. Finally, the cutting stress is related to 2045–2261 nm and 2554–2576 nm, which are characteristic wavelengths of polysaccharides based on C=O and C=C functional groups, respectively.

### 3.4. Predictive Performance of the Classification Model

The classification accuracy and performance results of each regression model are listed in Appendix A. Four classification models were designed: PLS-DA, VIP-PLS-DA, ANN-DA, and PLS-ANN-DA (PLS based on ANN-DA). VIP-PLS-DA is a new PLS model based on the critical wavelengths selected from the results of PLS-DA, and PLS-ANN-DA is the ANN model retrieved after PLS training.

For moisture, the classification accuracy for both the trained and predicted models was perfect, with R^2^ values measuring between 90.4% and 99.0%. However, VIP-PLS-DA showed relatively lower R^2^ values, ranging from 80.3 to 85.3%. The trained model achieved a classification accuracy of 98.0–100.0% for crude protein, with R^2^ values between 85.5% and 100.0%. However, in the predicted model, the classification accuracy of classes 1 and 2 ranged from 81.1 to 95.8%, with VIP-PLS-DA showing R^2^ values of 39.0% for class 1 and 53.6% for class 2. The PLS-DA and VIP-PLS-DA models achieved better results for cutting stress, with accuracies of 99.0–100.0% and R^2^ values between 76.1% and 94.4%. However, the ANN-DA and PLS-ANN-DA models showed a relatively lower accuracy (between 80.8% and 95.6%).

R^2^ and root-mean-square error (RMSE) values serve as indicators to confirm the performance of statistical models. A high RMSE value in the classification model indicates a relatively low accuracy and stability [55]. For industrial classification model development, an RMSE < 0.2 is required [56]. Models with R^2^ values > 70% can be considered significantly correlated [57]. Some cross-validation models in the PLS-DA showed RMSE values > 0.2. Nonetheless, the overall classification accuracy and performance of PLS-DA were higher than those of ANN-DA, making PLS-DA an appropriate regression model for developing a nondestructive quality classification system for dried laver.

### 3.5. Quality Grading

Tukey’s ANOVA-based classification of dried laver using a single quality factor (rating results for each quality factor) proved to be successful. Consequently, the present study aimed to establish a comprehensive standard for classifying the quality of dried laver through PLS-DA. Moisture content was classified as follows: it received a first-grade rating if it was <12% and a third-grade rating if it was >12%. Protein content was rated as first grade if it exceeded 35%, second grade if it was above 30%, and third grade if it fell below 30%. Moisture and protein content were rated according to the Korean Industrial Standards (KS H 6025). The texture of dried laver was categorized into three levels based on previous research, which indicates that tougher textures improve overall sensory preference [58]. Textures were classified as follows: if the cutting stress exceeded 0.45 kg·mm, it received a first-grade rating; if it exceeded 0.33 kg·mm, it received a second-grade rating; and if it was <0.33 kg·mm, it was assigned a third-grade rating.

Complex quality criteria based on the content of quality indices (moisture, crude protein, and cutting stress) for PLS-DA classification are shown in Table 1. Samples were located according to these complex criteria (Figure 5).

The samples were divided into two zones. Each location of the sample was determined by the mean value of the three quality indices ratings assigned to it according to the results of the physicochemical composition and cutting stress analyses. If the mean value of the rating was close to 1, it was assigned to zone 1, and if it was close to 2, it was assigned to zone 2. Zone 1 included samples with first-grade crude protein and first- and second-grade cutting stress. Zone 2 included samples with third-grade cutting stress or both second-grade crude protein and cutting stress. The samples that was present in zone 1 but assigned the × symbol was excluded from zone 1 because it exceeded 12% moisture and were instead assigned to zone 2.

### 3.6. Classification Through Quality Grading

The accuracy and performance of the quality grade classification are presented in Table 2 and Figure 6. The quality learning and predictive accuracy of dried laver classified according to Table 1 ranged from 96 to 100% in both the trained and predicted models, with cross-validation also showing a high accuracy of 96%. The HSI system effectively classified exceptions caused by moisture in zone 1 as zone 2. Therefore, HSI can reveal invisible physicochemical differences in each sample and classify them effectively using PLS-DA.

## 4. Discussion

Researchers have developed HSI systems that create 3D data by line-scanning samples to overcome this limitation. This method combines a 2D image of the sample with the wavelength data of each image pixel, allowing the prediction of physical and chemical component content and quality evaluation. The HSI system predicts sample content or classifies quality by analyzing the correlation between hyperspectral image wavelengths and quality factor data. Consequently, selecting the main wavelength for each quality factor is crucial for accurate classification.

This study assessed quality indicators for dried laver, such as moisture, protein content, and texture, using cutting stress. Each of these indicators showed significant effects in independent areas. For instance, the moisture content of various plants, such as red ginseng, ginger, persimmon, and radish, can be measured within the 1396–1468 nm range; similarly, the moisture content of dried laver was confirmed in this wavelength range [59,60]. Protein content was measured at the same wavelengths as those of white maize kernel protein, within approximately the 2000 nm region [61]. Depending on the amino acids, proteins, and analysis techniques, the main protein wavelength range can extend to 2500 nm [54].

For texture evaluation, various VIS/NIR and SWIR wavelength bands (400–2500 nm) have been used to assess the softness of meat and fish and the hardness of fruits and vegetables [20]. The texture of food products is determined by the density of dietary fiber and protein [62,63]. Wavelengths between 2045 and 2261 nm and between 2554 and 2576 nm were selected as critical indicators for classifying the texture of dried laver. The texture of dried laver was further predicted using the 2090–2110 nm and 2273–2297 nm regions, which were identified as primary absorption wavelengths for snap bean fibers and cellulose [64]. This suggests that spectral absorption as a function of physicochemical composition is the same for laver.

After selecting the relevant wavelengths, three preprocessing stages were required: sample range selection, ROI identification through clustering, and wavelength calculation. The ROI can be specified on a pixel basis or by using a deep learning approach [65]. The isodata clustering method used in this study minimized clustering cases for the hyperspectral image processing of dried laver, with an average porous area of 2.94%.

PLS-DA, which was utilized in this study, is known for its high classification accuracy across multiple samples [66]. All four quality prediction methods, including PLS-DA, ANN-DA, and PLS-ANN-DA, showed promising accuracies, each exceeding 80%. However, in the case of VIP-PLS-DA, the coefficient of determination significantly decreased, and the RMSE increased when classifying the three quality indicators. Furthermore, the classification performance of ANN-DA and PLS-ANN-DA also declined. The finding that simple PLS-DA outperforms both ANN-DA and VIP score-based PLS-DA in food quality classification aligns with previous research, such as studies on seed variety classification and citrus spoilage prediction [67,68].

The classification accuracy of individual quality indicators, such as moisture, protein, and cutting power, using PLS-DA ranged from 81% to 100%. Notably, the accuracy of the quality classification achieved with the dried laver quality classification system presented in this study was 100%. This high level of accuracy with PLS-DA has been reported in numerous previous studies. For instance, in evaluating the fertilizer suitability for tea plants, a PLS classification accuracy of 100% was observed [69]. Additionally, the prediction accuracies for protein and starch contents in corn were reported to be 80% and 79%, respectively [70]. The ability of PLS-DA to demonstrate high classification accuracy based not only on single quality factors but also on a comprehensive grading system that incorporates various quality indicators highlights the potential scalability of food quality classification technology.

In evaluating the classification results of quality indicators using PLS-DA, we observed that the coefficient of determination for the validation set was relatively low, at 39.8% to 58.4%, despite a high overall accuracy. This suggests limited explanatory power between the quality indicators of dried laver and the quality classification system. It is important to note that the classification system used in this study is a multinomial logistic regression that categorizes data into three groups. As a result, the correlation between the variance in the group data and the model fit appears weak [71].

Despite having a low coefficient of determination, the RMSE of the PLS-DA model was still below 0.3. Additionally, the classification results based on the quality rating system (as shown in Table 2) also maintain an RMSE below 0.3. Determining an appropriate RMSE in the data prediction process can be challenging, as it depends on the variance and characteristics of the dataset. However, in studies focused on developing prediction models for fields such as medicine, pharmaceuticals, and food, an RMSE of 0.3 indicates sufficiently high prediction performance [72,73,74]. Therefore, we conclude that PLS-DA is robust enough to be applied to a quality grading system. Nonetheless, there is potential for improved classification performance and higher coefficients of determination by incorporating more advanced algorithmic models, such as Convolutional Neural Networks, or by using major wavelength selection techniques beyond the VIP score [75].

The quality grading system for dried laver, utilizing HSI, is a promising approach. This technology can objectively assess the quality and safety of each dried seaweed product. However, there have been very few instances where HSI has been applied to evaluate food classification performance in relation to quality grading systems. Although HSI research has successfully detected foreign substances in food production and processing [76] or classified areas within ROIs based on physicochemical composition ratios [77], these applications are not yet integrated with the grading systems commonly used in the industry, which presents limitations.

This scarcity of examples hinders the evaluation of quality classification techniques using HIS. Consequently, research on food quality grading systems employing HIS should continue.

## 5. Conclusions

This study investigated physicochemical and sensory quality indices to develop a comprehensive quality grading system for dried laver. The results demonstrated that each quality index could be successfully classified using PLS-DA, linear regression models, and ANN-DA machine learning-based regression, achieving a predictive accuracy of over 80%. Critical wavelengths for classifying each quality index were identified, enabling the proposal of numerical and objective criteria for grading the quality of dried laver. Our findings could enhance traditional quality standards by providing precise values, thereby saving time and resources while improving the accuracy and efficiency of quality assessments.

## Figures and Tables

**Figure 1 foods-14-00497-f001:**
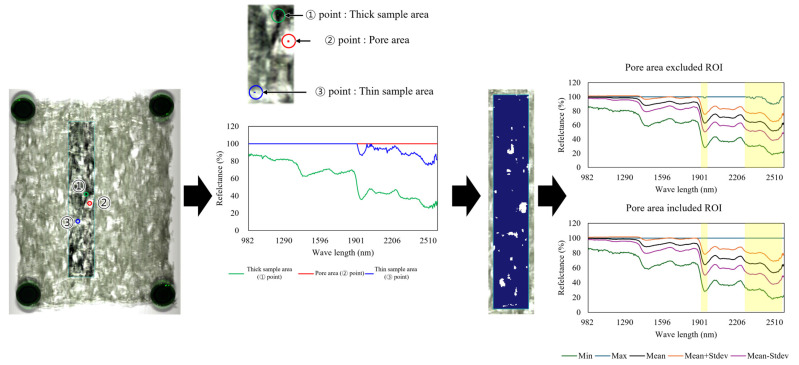
Scheme illustrating the process of excluding a pore area in the ROI spectrum of dried laver. ROI: region of interest.

**Figure 2 foods-14-00497-f002:**
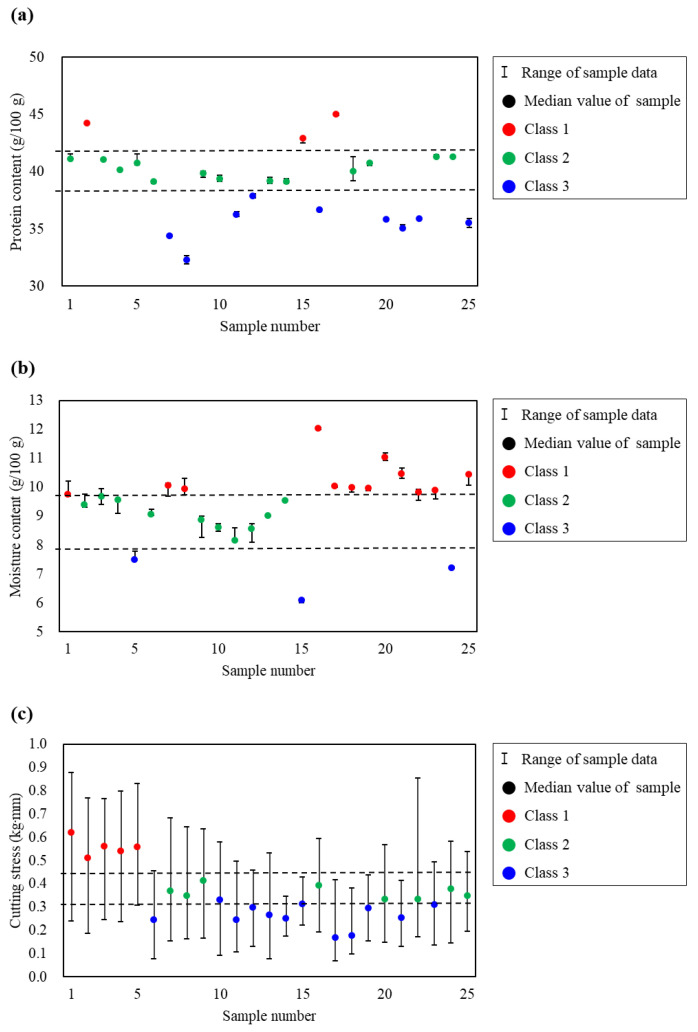
Quality index analysis and classification results based on Tukey’s analysis of variance test. (**a**) Moisture, (**b**) crude protein, and (**c**) cutting stress.

**Figure 3 foods-14-00497-f003:**
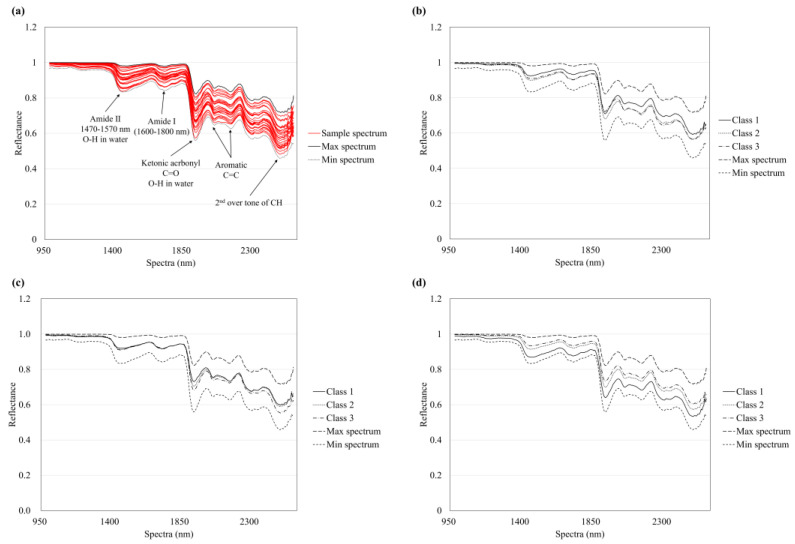
Reflectance measured at each wavelength for dried laver samples and that measured at each wavelength for each class of quality indices based on Tukey’s analysis of variance. (**a**) Whole sample, (**b**) moisture, (**c**) crude protein, and (**d**) cutting stress.

**Figure 4 foods-14-00497-f004:**
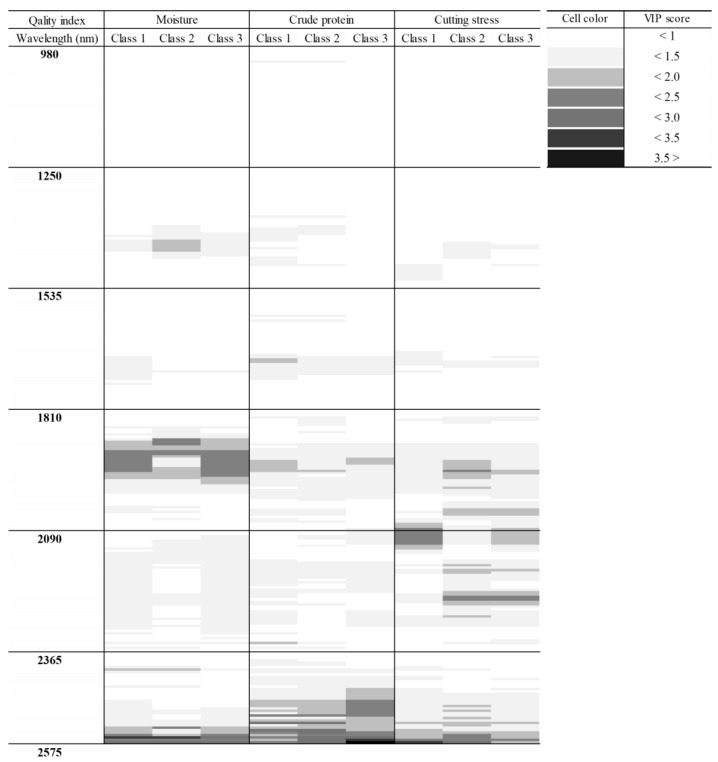
Wavelength selection for classifying quality indices using variable importance in projection scores based on Tukey’s analysis of variance.

**Figure 5 foods-14-00497-f005:**
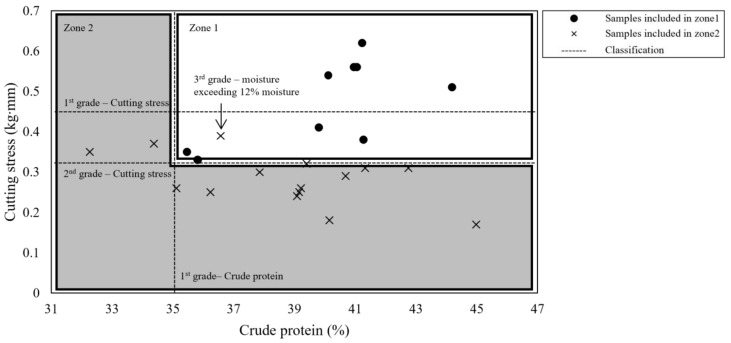
Results of the dried laver classification grading, as evaluated using quality criteria.

**Figure 6 foods-14-00497-f006:**
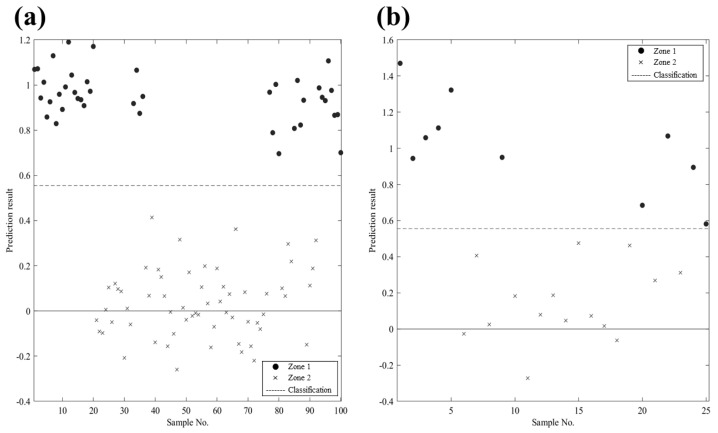
Trained (**a**) and predicted model (**b**) results of PLS-DA based on the hyperspectral imaging wavelength of the dried laver classification grading, as evaluated using quality criteria.

**Table 1 foods-14-00497-t001:** Standard values of the quality index for the quality grading system.

Quality Index	Classification Criteria		
	First Grade	Second Grade	Third Grade
Moisture	<12% ^(1)^		>12%
Crude protein	≥35%	≥30%	<30%
Cutting stress	≥0.45 kg∙mm	≥0.33 kg∙mm	<0.33 kg∙mm

^(1)^ Assigned first grade if the moisture content is ≤12%; there was no classifying second-grade.

**Table 2 foods-14-00497-t002:** Classification performance of dried laver samples using optimal criteria in the quality grading system.

	Accuracy			R ^(2,4)^			RSME ^(5)^		
	TM ^(1)^	CV ^(2)^	PR ^(3)^	TM	CV	PR	TM	CV	PR
Classification performance	100%	96%	100%	92%	75%	78%	0.14	0.25	0.25

^(1)^ Trained model; ^(2)^ cross-validation model; ^(3)^ predicted model; ^(4)^ coefficient of determination; ^(5)^ root-mean-square error.

## Data Availability

The original contributions presented in this study are included in this article and Appendix A. Further inquiries can be directed to the corresponding author.

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
