# Peer review of "Short-Wave Infrared Hyperspectral Image-Based Quality Grading of Dried Laver (Pyropia spp.)"

_foods, 2025, doi:10.3390/foods14030497_

Round 1

Reviewer 1 Report

Comments and Suggestions for Authors

#Review

·      Authors names: delete the word “by”.

·      Introduction section comments.

·      Why is dried laver of great concern, is it that the fresh or raw laver is unaccepted by buyers or has some nutritional defect compared to dried laver. Authors should clearly give a justification to the immediate drying process fresh laver is subjected to other than the statement “raw laver quickly losses its freshness”. Authors should establish the clear food nutrient / human nutritional fact that is associated with the quick loss of laver freshness. Is profit reduced due to buyers’ willingness to buy fresh quality laver? Or raw laver lacks some nutritional qualities, and its dry form help in extending the shelf-life? Although, the quality of dried laver is likely related to its protein content and texture point is clearly established but there must be something peculiar to the consideration of choosing dried laver over raw/fresh laver from a scientific, traditional, and nutritional point of view accustomed to Korean cuisine.

·      Also is the dried laver demand particularly in South Korea or different in other nations like Japan and China.

·      Kindly crosscheck and consider changing the better the use of “raw” and “fresh” laver to for better coherency of the later drying process used to make the presentation more scientific.

·      “Recently, HSI has been utilized for classification using multivariate regression algorithms [30].” Statement is incomplete. Kindly rewrite to complete it. HIS was used to classify what strawberry, meat, apple, which foo product?

·      In general, the introduction was well structured and written. Also, the objectives of the study were clearly presented devoid of discrepancies.

·      Materials and methods section: define HPLC and AOAC and keep abbreviation in parenthesis.

·      Figure 1: is it possible for authors to change the legend colors to different colors individually for the corresponding indices. For example, it seems Min and Max has red colors, whiles Mean+Sidev both has green colors. Or the red color for Min or Max are for “pore are excluded” and “pore are included” figures. I suggest authors consideration for different specific colors of legends would perfectly correspond to their respective index to avoid discrepancies. Kindly revise that section of the figure.

·      Also, the description of figure 1: “ROI, region of interest” must be written as “ROI: region of interest”.

·      “Sensitivity is defined as the ability of the model to correctly recognize samples belonging to the target class”. Citation needed.

·      “The specificity is de-fined as the ability of the model to correctly recognize samples belonging to a different class”. Citation needed.

·      Figure 3: kindly improve the resolution of the figure since the present form looks blurry. Or its resolution quality has been reduced by the upload of the manuscript and its conversion process?

·      Table 1: “<12%” falls under which column of classification criteria (1st grade, 2nd grade, and 3rd grade). Kindly correct it.

·      Figure 5: check and enhance figure resolution. Also, kindly define the legends (x, ·) beneath the figure and clearly give a description of the figure for coherency. The figure seems vague and confusing considering the nature white and ash color portions are define coupled with the positioning of the respective zones and grades.

·      Table 3: define abbreviations TM, CV, R2, below the table.

·      The 4th, 5th and 6th paragraph of the discussion section lacks a comparative analysis of present findings with other studies existing results as presented in same section paragraph 2 and 3. It would be of great approach to enrich the section with inferences from similar existing works to make the presentation of findings more scientific. Again, if the findings turn to be new without a better inference to existing literature you kindly consider establishing the fact that it is a maiden finding.

Author Response

Thank you for considering my article for publication in Foods.

I am grateful to you and all the reviewers for the valuable suggestions provided.

Below are the responses to the reviewer’s comments.

Reviewer 2 Report

Comments and Suggestions for Authors

This manuscript explores the application of short-wave infrared hyperspectral imaging (SWIR HSI) for quality grading of dried laver (Pyropia spp.), a commercially important seaweed. The authors investigate the relationship between spectral data and key quality indices, including moisture, protein content, and cutting stress. The use of HSI for rapid and objective quality assessment is a relevant and timely topic, offering potential advantages over traditional methods. The study employs partial least squares discriminant analysis (PLS-DA) and compares its performance with other methods for spectral data analysis. While the premise is promising and the results generally positive, the manuscript requires substantial revisions to improve clarity, rigor, and overall impact. Specifically, the introduction needs strengthening to better contextualize the study within the broader field of food quality assessment and highlight the novelty of the approach. The methodology section requires more detail to ensure reproducibility, and the discussion should be expanded to provide a more in-depth interpretation of the findings and their implications. Furthermore, the manuscript suffers from several stylistic and grammatical issues that need to be addressed.

Specific comments and suggestions for improvement:

Introduction: The introduction provides a reasonable background on laver and its importance, but it lacks a strong justification for the specific use of SWIR HSI. Expand on the limitations of existing quality assessment methods and clearly articulate the advantages of the proposed HSI approach. Highlight the novelty of the study and its potential contribution to the field. The transition between paragraphs could be smoother. For example, the connection between the health benefits of laver and the need for quality grading is not explicitly stated. More recent and relevant references could be included to strengthen the background information and highlight the current state of research in this area. The introduction should also clearly state the objectives of the study.

Materials and Methods: This section requires significant improvement in detail and clarity. Provide more specific information about the laver samples (species, origin, processing methods). The description of the HSI system setup is insufficient. Specify the camera model, lens specifications, illumination source, and data acquisition parameters in more detail. Justify the choice of the SWIR range. The explanation of the ROI selection and processing is unclear. Provide a more detailed description of the isodata classification method and its parameters. Clarify the rationale for selecting a square ROI and its specific dimensions. The data analysis section needs more detail on the implementation of PLS-DA, VIP score calculation, and ANN-DA. Specify the software used, pre-processing steps, and model validation procedures. Define all abbreviations at first use (e.g., VIP, RMSE).

Results: The discussion of spectral characteristics is rather superficial. Provide a more in-depth analysis of the spectral features and their relationship to the chemical composition of laver. The rationale for selecting specific wavelengths needs to be more clearly explained and justified. The discussion of the predictive performance of the classification models is difficult to follow. Present the results in a more organized and concise manner. Clearly explain the meaning and significance of the different performance metrics (accuracy, R², RMSE). The justification for selecting PLS-DA as the optimal model is not entirely convincing.

Discussion: This section is too brief and lacks depth. Expand on the interpretation of the findings and their implications for quality grading of dried laver. Discuss the advantages and limitations of the proposed HSI approach compared to other methods. Connect the results to the broader context of food quality and safety. Address the potential practical applications of the developed grading system.

Author Response

(The authors gave the same response as above.)

Reviewer 3 Report

Comments and Suggestions for Authors

The manuscript presents a study for quality grading of dried laver using short wave infrared hyperspectral image. The HPI method offers a rapid and accurate quality assessment of moisture content, protein content, and cutting stress of dried laver. The PLS-DA regression model demonstrated high accuracy by analyzing the spectral signatures. Some issues need to be addressed.

1) In the introduction, it mentioned that the quality and flavor of dried laver is likely related to its protein content. Is there evidence to show the direction correlation between these two? So the protein content can be used to reflect the flavor quality of laver.

2) The cost of HSI method should be discussed. Because normally the HSI instrument is quite expensive. Is it cost-effective to use HSI method for dried laver?

3) How many data points were used for modeling training and how many data points were used for validation? In the manuscript, it mentioned 25 samples were analyzed. It normally needs a large dataset to establish an accurate model.

Author Response

(The authors gave the same response as above.)
